# Interactive Analysis of Functional Residues in Protein Families

Morgan N. Price,[a] Adam P. Arkin[a,b]

aEnvironmental Genomics and Systems Biology, Lawrence Berkeley National Laboratory, Berkeley, California, USA
bDepartment of Bioengineering, University of California, Berkeley, California, USA

**ABSTRACT** A protein's function depends on functional residues that determine its binding specificity or its catalytic activity, but these residues are typically not considered when annotating a protein's function. To help biologists investigate the functional residues of proteins, we developed two interactive web-based tools, SitesBLAST and Sites on a Tree. Given a protein sequence, SitesBLAST finds homologs that have known functional residues and shows whether the functional residues are conserved. Sites on a Tree shows how functional residues vary across a protein family by showing them on a phylogenetic tree. These tools are available at http://papers.genomics.lbl.gov/sites.

**IMPORTANCE** For most microbes of interest, a genome sequence is available, but the function of its proteins is not known. Instead, proteins' functions are predicted from their similarity to other protein sequences. Within a protein's sequence, a few key residues are most important for function, such as catalyzing a chemical reaction or determining what it binds. But most function prediction tools do not take these key residues into account. We developed interactive tools for identifying functional residues in a protein sequence by comparing it to proteins with known functional residues. Our tools also make it easy to compare key residues across many similar proteins. This should help biologists check if a protein's function is predicted correctly, or to predict if groups of similar proteins have conserved functions.

**KEYWORDS** functional residues, protein sequence analysis

A protein can be thought of as a three-dimensional scaffold that places key functional residues, for binding or for catalysis, in the correct locations. Although these functional residues are critical for proteins' functions, they are not considered by most of the widely used tools for automatically annotating protein functions (1–4). The only exception that we are aware of is UniProt's UniRule, which records the active site residues for many protein families and adds a warning if any of them are altered (5).

If a protein is of particular interest, then manual analysis of its functional residues, as inferred from experimental studies of homologous proteins, is more effective than an automated approach. Specifically, a human analyst can often find information in research articles or protein structures that is not represented in the annotation databases. However, doing these manual analyses is laborious. First, it is not obvious which homologs have experimental data about their functional residues. Second, once experimental data about key residues in a homolog is found, it can be quite cumbersome to identify the corresponding residue in the protein of interest. For proteins with structures, there's usually two different residue numberings, corresponding to the natural sequence and the portion of the sequence that was crystallized and whose structure was resolved; when reading a manuscript, it is not always obvious which coordinate system is being used. If multiple members of a family have been studied, a manuscript may use residue numbers from a reference protein instead of from the protein being studied. And key residues are sometimes shown as highlighted columns in alignments, but without residue numbers.

If the functional residues are partially conserved, it is helpful to see how those residues vary across the family. This is particularly useful for similar proteins with known function,

Address correspondence to Morgan N. Price, funwithwords26@gmail.com.

The authors declare no conflict of interest.

as this can reveal if a change to a functional residue is likely to lead to a change in function. However, functional residues are often far apart in the sequence, so alignment viewers do not make it easy to view functional residues across a protein family.

To make it easier to examine the functional residues of a protein or a protein family, we developed SitesBLAST and Sites on a Tree. SitesBLAST compares a protein of interest to a large database of proteins with known functional residues. Sites on a Tree shows key residues across a protein family, along with a phylogenetic tree to show how the sequences are related to each other.

## RESULTS

**Experimentally identified functional residues for 100,000 proteins.** As of April 2022, SitesBLAST's database includes functional residues for 125,195 distinct protein sequences. SitesBLAST relies on two sources of functional residues: BioLiP (6) and Swiss-Prot. BioLiP incorporates protein-ligand interactions from protein structures in the Protein Data Bank (PDB), with biologically irrelevant ligands removed. BioLiP also includes active site residues if they are annotated in the PDB entry. In SitesBLAST's database, 94,655 distinct protein sequences have information from BioLiP. Most of the functional residues from BioLiP (97%) are involved in binding.

Swiss-Prot is the curated subset of UniProt (7) and includes many kinds of "sequence features," along with evidence codes. SitesBLAST's database only incorporates features from Swiss-Prot if they have experimental evidence. Many of the features from Swiss-Prot (45%) indicate a covalent modification to the protein. These are included in SitesBLAST's database because they are often important for a protein's function, but strictly speaking, many of them are not functional residues. Another 38% of the features from Swiss-Prot describe experimentally mutated residues. These are annotated regardless of whether mutating the residue had an effect, so not all the mutated residues are important for function. If modified and mutated sites are ignored, then the number of distinct protein sequences from Swiss-Prot with functional sites drops from 31,131 to 8,442.

**SitesBLAST.** At the SitesBLAST website, you can enter a protein's sequence or identifier. SitesBLAST will compare the query to its database, using protein BLAST, and will show up to 20 alignments. For example, as shown in Fig. 1, the alternative homoserine kinase BT2402 is similar to the B chain from a crystal structure of a phosphoglycerate mutase. The structure includes two zinc ions and a calcium ion. As shown at the bottom of Fig. 1, the zinc binding sites are fully conserved (for instance, D12 aligns to D9 in BT2402). In contrast, the calcium binding site is not conserved (i.e., E41 versus R38).

To make it easier to relate the binding sites to the alignment, SitesBLAST has interactive highlights. Hovering on a binding site at the bottom such as "E41 ($\neq$ R38)" will highlight the corresponding location in the alignment (black box in the top row of Fig. 1). Conversely, hovering on a functional residue in an alignment will highlight the same site at the bottom (not shown). Also, hovering on any alignment position will show the residue numbering in both sequences. Overall, SitesBLAST takes just a few seconds to highlight potential functional residues, and whether they are conserved.

**The coverage of SitesBLAST's database.** To estimate the coverage of SitesBLAST, we selected 1,000 proteins at random from UniProt's reference proteomes and compared them to SitesBLAST's database using protein BLAST. A total of 56% of the queries had hits (E $\leq 10^{-3}$), and 49% had hits with 30% identity or higher. Homologs at 30% identity or higher are likely to have similar functions, and the alignments are likely to be accurate. We checked a random sample of 40 of the proteins with hits of at least 30% identity. All but one of these had functionally informative hits with known active site residues, residues that bind to substrates or other biologically relevant ligands, or residues whose mutation leads to a loss of function. The final case (UniProt: A0A2T6DQW1) was ambiguous: there are protein structures of homologs in complex with inhibitors or with ligands whose biological relevance is not proven. Overall, given a random protein that was predicted from a genome's sequence, SitesBLAST can identify potential functional residues about half of the time.

We also compared the coverage of SitesBLAST to that of the Conserved Domains Database (CDD) (8). CDD includes functional sites for many of its families, and the CD-Search

## 2zktB Structure of ph0037 protein from pyrococcus horikoshii

34% identity, 99% coverage: 2:405/407 of query aligns to 5:374/381 of 2zktB

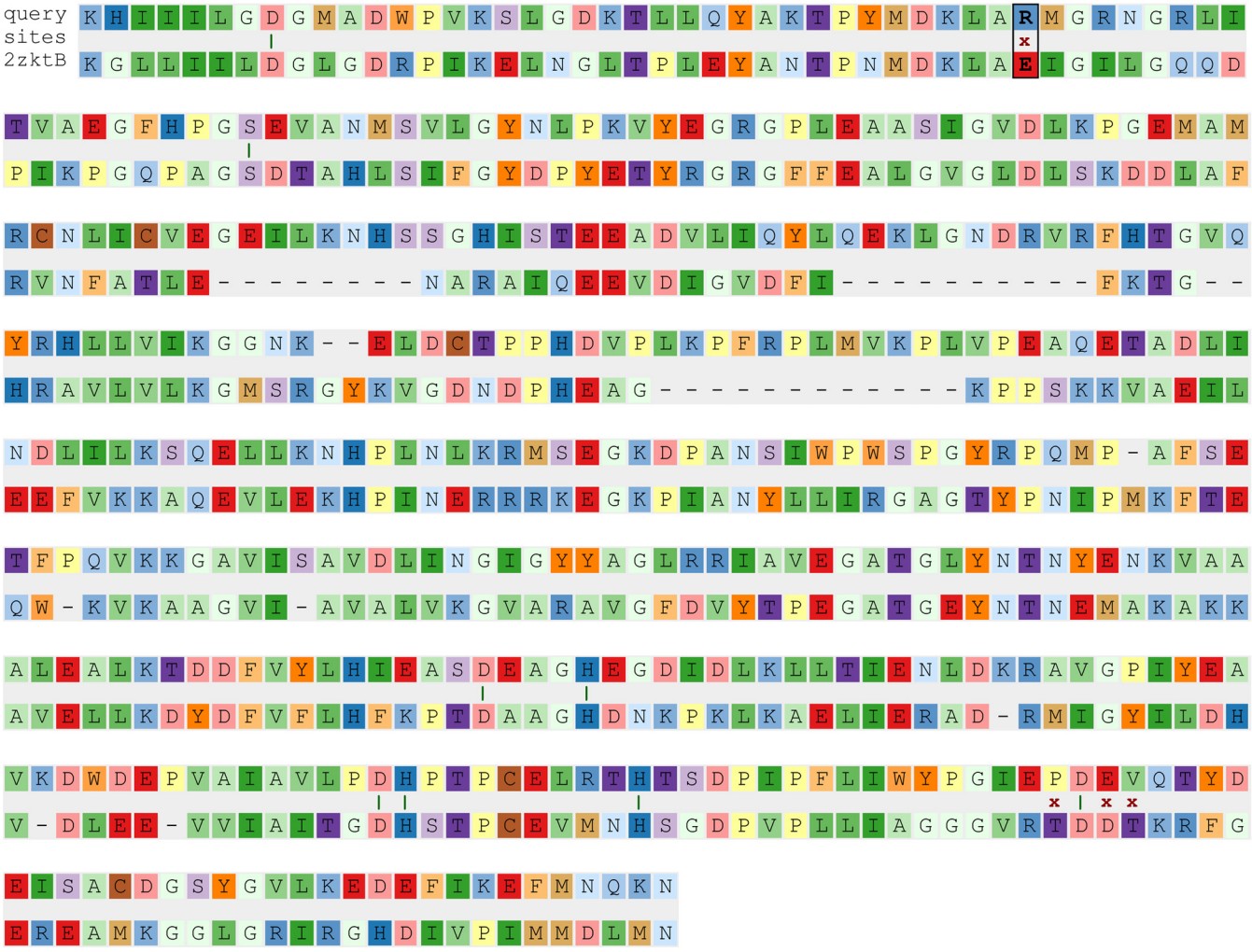

- binding calcium ion: E41 (≠ R38), T341 (≠ P372), D342 (= D373), D343 (≠ E374), T344 (≠ V375)
- binding zinc ion: D12 (= D9), S59 (= S56), D274 (= D302), H278 (= H306), D315 (= D346), H316 (= H347), H325 (= H356)

**FIG 1** An example alignment from the SitesBLAST website. Each aligned functional residue is highlighted with a dark green vertical bar if the two sequences match, or with a red x otherwise.

web tool can highlight functional positions in the alignment of a query sequence to a homologous family. When we ran CD-Search on our test set of 1,000 proteins, it reported site information for 40% of the proteins, which is lower than the coverage of SitesBLAST (49%). Most of the sequences with sites from CDD had hits in SitesBLAST as well (87% if we disregard the 30% identity filter).

**Sites on a Tree.** Where SitesBLAST compares two sequences at a time, Sites on a Tree shows multiple sequences in a family. When considering how functional residues vary within a family and determine a protein's function, the most informative sequences are for proteins whose function is known. So, given a protein of interest, Sites on a Tree can identify homologs that have known functional sites (as in SitesBLAST) or whose function is known. The analyst can also add other proteins of interest to the list. Given these proteins, the website builds an alignment with MUSCLE 3 (9) and infers a phylogenetic tree with FastTree 2 (10). Each of these steps usually takes a few seconds. Alternatively, the analyst can perform any of these steps themselves and upload unaligned sequences, an alignment, or a tree. Sites

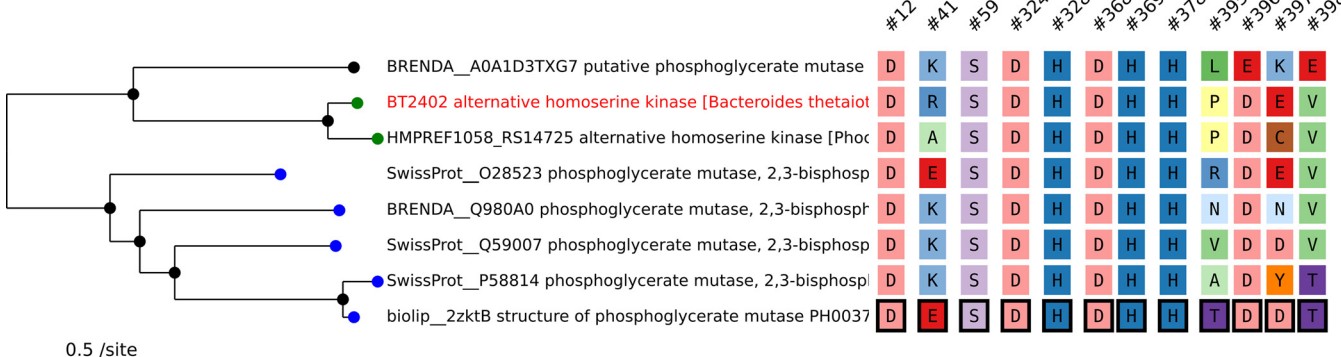

**FIG 2** Sites on a Tree results for the alternative homoserine kinase BT2402. We used Sites on a Tree to automatically select characterized homologs, build a tree and alignment, and select known functional sites (highlighted with boxes). Other positions in the alignment are omitted. Each functional residue has hover text to describe its role. In the tree, leaf nodes are color coded by the protein's function.

on a Tree supports sequences in fasta format, alignments in fasta, clustal, or Stockholm format, and trees in newick format.

Sites on a Tree can show the known sites (from SitesBLAST's database), or the analyst can choose which sites to show. As shown in Fig. 2, when showing known sites, SitesBLAST highlights them with boxes. Sites on a Tree can identify known sites in uploaded sequences if, after removing gaps, the sequence is identical to a sequence in SitesBLAST's database.

Alternatively, the analyst can choose which residues to show by entering alignment positions or positions within the "anchor" sequence; by default, the original query is the anchor. To help the analyst find the correct residue number, Sites on a Tree can list all of the matches, across all of the sequences in the alignment, for subsequences or patterns such as NSG, CxxC, or DEA[DH].

The analyst can also customize the view. For example, in Fig. 2, sequences with the same function have the same coloring in the tree; these colors were set by uploading a table with a color for each protein identifier. The uploaded table can also contain a description and a web link for each protein. Alternatively, there's a link to download the tree+sites graphic in scalable vector graphics (SVG) format, which can be edited in tools such as Inkscape or Adobe Illustrator.

**Visualizing functional sites across hundreds of sequences.** The tree+sites graphic (such as shown in Fig. 2) works well for up to a few dozen sequences, but what if the analyst is studying a large family? If the analyst has chosen which residues to include, Sites on a Tree shows a more compact view (Fig. 3). If a sequence position is conserved across a clade in a tree, then the amino acid code is drawn just once. If a single sequence or a small clade has a variant residue, and the clade is too small to draw the amino acid code, then only the color is shown. The compact view is only used when the analyst is choosing which sites to show, as there is no way to highlight the known functional residues in a specific sequence in the compact view.

To ensure that variant residues can be seen in the compact view, Sites on a Tree needs to use a different color for every amino acid. Unfortunately, none of the standard color schemes for amino acids do this. We used the RColorBrewer library to select 12 paired colors and interpolated within each pair to get additional colors for each group of similar amino acids. The groups are negatively charged residues (DE); small polar residues (ST); positively charged residues (NQKRH); aromatic residues (FWY); small hydrophobic residues (GAVLI); and other residues (PMC). Gaps in the alignment are shown in gray. For the larger groups (NQKRH and GAVLI), we altered the lighter color at the end of the scale, to make them easier to discriminate. Despite our best efforts, it can still be difficult to identify a residue by its color alone, but Sites on a Tree provides mouseover text and a zoom feature. Clicking on an internal node in the tree will navigate to a new view for just that clade with enough vertical space to label each variant residue.

If there are more than 30 sequences, then the compact view does not have space for the protein's identifiers. Instead, clicking on a leaf node will show the protein's identifier. Then, hovering on the identifier will show the description, and clicking on it will navigate to

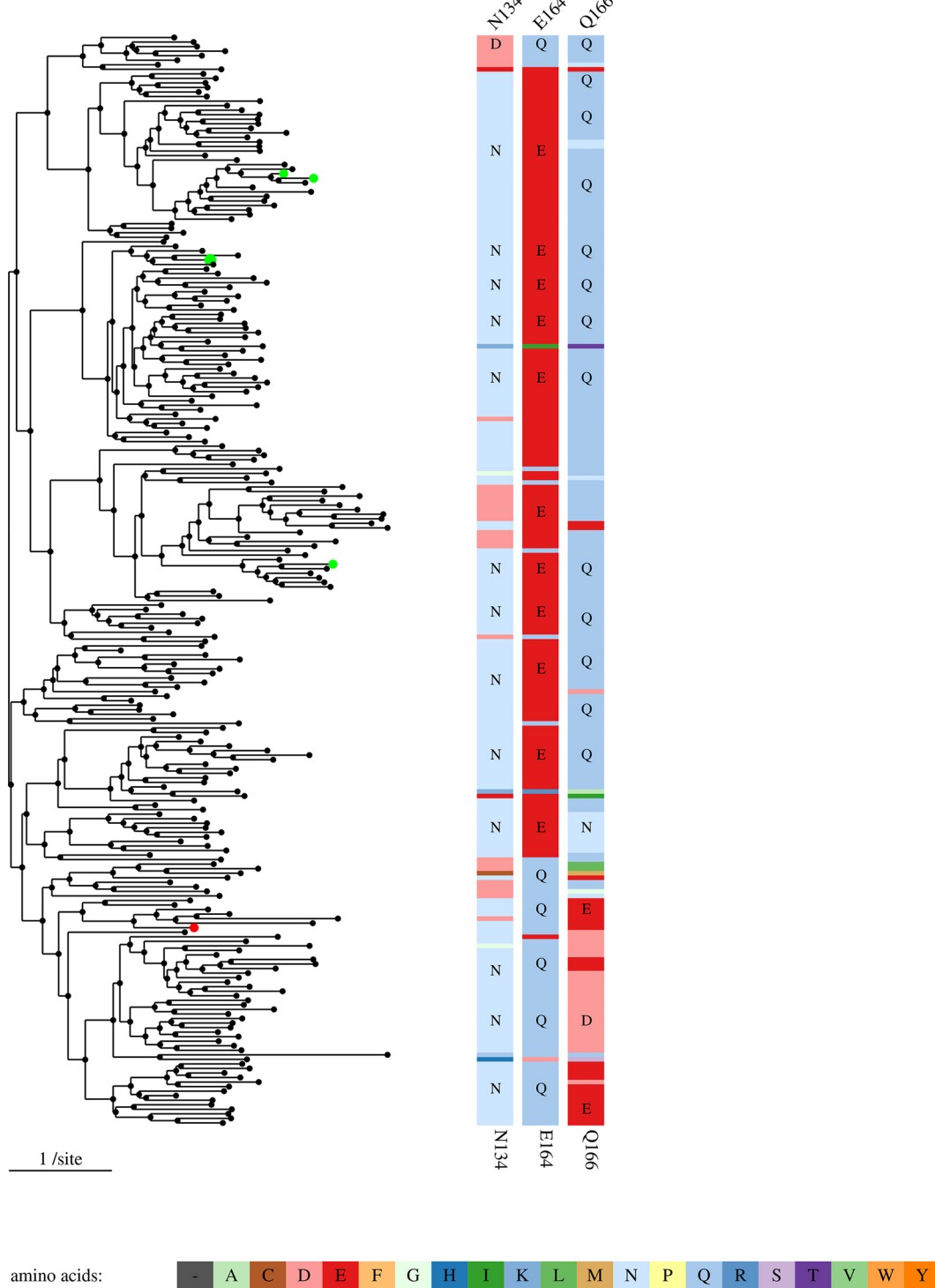

**FIG 3** Putative active site residues for 240 sequences from the 3-ketoglycoside hydrolase family. In this screenshot from Sites on a Tree, the left side shows a phylogenetic tree. Proteins of known function are highlighted by colored circles. The right side shows the alignment for three positions, with each amino acid coded by color (see legend at the bottom). Amino acid labels are shown if the residue is conserved across a large enough subtree. The label of each alignment position (at the top) is based on the anchor sequence (BT3761).

a page which includes sequences, a list of known functional sites (if any), and links to other sequence analysis tools. To locate family members of interest within the tree, Sites on a Tree has a search feature which highlights proteins whose identifiers or descriptions match the query. The search feature was inspired by the ATV viewer (11).

The example in Fig. 3 has 240 sequences, which is about as much as will fit in our computers' screens. For larger families, we recommend using CD-HIT or usearch (12, 13) to remove highly similar sequences and hence to reduce the size of the tree. In fact, for our example, we began with all the sequences from the 3-ketoglycoside hydrolase family from MicrobesOnline (14) that have an alignment score of 80 bits or more against PFam PF06439.11 (15). We added a few more sequences for characterized proteins. This gave 646 sequences, which in our experience takes 2 to 3 screenfulls if viewed in Sites on a Tree. We clustered these at 60% identity to get a more manageable visualization of 240 sequences (Fig. 3).

## DISCUSSION

SitesBLAST and Sites on a Tree are both ways of viewing functional sites. SitesBLAST is simpler and quicker to use than Sites on a Tree: just put in one sequence and see the pairwise alignments. However, in many situations, Sites on a Tree is more powerful. If there are several homologs with known functional sites, it can be easier to understand all these sites in the context of a multiple sequence alignment. Also, Sites on a Tree includes similar sequences of known function, even if they do not have any known functional sites. This can help the analyst decide if a change to a functional site is likely to imply a change in the protein's function. Or, if the analyst provides their own sequences, then Sites on a Tree can show selected sites for uncharacterized proteins. Sites on a Tree can visualize alignments with hundreds of sequences. Sites on a Tree may also be useful for visualizing putative functional residues that were identified by automated tools (16).

The most similar tools we are aware of are firestar, a structure-based tool for predicting functional residue (17), and the Conserved Domain Database (8). The extended results view of firestar is similar to SitesBLAST, but SitesBLAST takes a few seconds, while firestar takes several minutes. Also, SitesBLAST includes functional residues from both Swiss-Prot and protein structures, while firestar only includes functional residues from structures.

CDD records functional sites for some of its families and can highlight functional sites on the alignment of the query sequence to curated members of the family. We found that SitesBLAST had higher coverage than CDD, with sites for 49% of prokaryotic proteins instead of 40%. Our impression is that SitesBLAST's database is larger because BioLiP is semiautomatically updated as structures of proteins bound to ligands become available. On the other hand, because CDD uses position-specific weight matrices, it can find more distant homologs than SitesBLAST can. Another major difference is that CDD annotates sites on families, while SitesBLAST focuses on information about individual homologs. Focusing on individual homologs often gives more complicated results, but it is easier to trace the results to the underlying data. Also, in the CDD results, it is often not clear which of the members of the seed alignment are of known function, or even if any of them are. This can make it difficult to decide if a change in sequence is likely to lead to a change in function. This limitation can occur with SitesBLAST's results as well, if there is no paper describing the structure.

Regardless of which tools are used, there's often more knowledge about the functional residues in the papers than in the databases. Conversely, ligand binding sites in protein structures may not be important for function inside the cell. So, when using SitesBLAST, it is important to read the paper that describes the protein structure (if there is one). We also recommend looking for additional relevant papers, for instance using PaperBLAST, which finds papers about a protein and its homologs (18). However, our impression is that for many protein families, there is no experimental evidence as to their functional sites.

SitesBLAST and Sites on a Tree are tools for exploration; they won't necessarily indicate a protein's function. For enzymes, if all the key active-site and substrate-binding residues are conserved, then the function is probably conserved as well. But it is often difficult to be sure that all the key residues have been identified.

## MATERIALS AND METHODS

**Data sources.** Swiss-Prot and BioLiP were downloaded in April 2022. Sites on a Tree also uses a database of over 100,000 characterized proteins, taken from the characterized subset of the PaperBLAST database (19); we used the April 2022 release. UniProt reference proteomes were downloaded in May 2020.

**Swiss-Prot sequence features.** Swiss-Prot describes many different types of sequence features and not all of them are included in SitesBLAST's database. For protein modifications, we used the CARBOHYD, CHAIN, CONFLICT, CROSSLNK, DISULFID, INIT_MET, LIPID, MOD_RES, NON_CONS, NON_STD, PEPTIDE, PROPEP, SIGNAL, TRANSIT, UNSURE, VAR_SEQ, and VARIANT features. But VARIANT features were ignored if the feature comment contains only a gene name, a strain name or a dbSNP reference. For binding, we used the BINDING, CA_BIND, DNA_BIND, METAL, and NP_BIND features. For other functional features, we used ACT_SITE, MOTIF, REGION, and SITE features. MUTAGENESIS features were stored as a separate category. Sequence features of any type were only included if they were based on experimental evidence (evidence code ECO:0000269 https://evidenceontology.org/term/ECO:0000269/).

**Phylogenetic trees.** When inferring a phylogenetic tree, Sites on a Tree trims the alignment to remove columns that are ≥50% gaps or that have more lower-case than upper-case letters. HMMer's hmmalign uses lower case for positions that are not actually homologous (http://hmmer.org/). Either of these trimming steps can be disabled. The trimmed alignment is used to infer the phylogenetic tree but is not used elsewhere (the site only shows the untrimmed alignment). After inferring a tree with FastTree 2, Sites on a Tree uses midpoint rooting to select the root of the tree.

If the analyst uploads a tree, then Sites on a Tree will treat the tree as rooted. Note that most tree inference tools produce unrooted trees, with the tree represented with an arbitrary root. In a fully resolved tree, the root node has two children if the tree is rooted and three if the tree is unrooted; Sites on a Tree allows multifurcations, so it will accept either type. Trees can be rerooted with tree editors such as FigTree, MEGA4, or phylip's retree.

**Software and software settings.** SitesBLAST's database is stored using sqlite3 (in the same database file as PaperBLAST's database) and as a protein BLAST database. SitesBLAST and Sites on a Tree are implemented in Perl (version 5.16.3) and HTML 5. Sites on a Tree uses JavaScript for interactive highlighting. For Sites on a Tree, the tree and the aligned residues are rendered using SVG.

SitesBLAST uses protein BLAST (version 2.2.18) with an E value cutoff of 0.001 and filters the query sequence for lookup only (-F "m S"). Sites on a Tree uses the same settings but only reports homologs that cover at least 70% of the query. Also, by default, Sites on a Tree only includes homologs in alignments if they are at least 30% identical to the query. The identity cutoff ensures that sequence alignments are likely to be accurate. Also, more distantly related sequences may have unrelated functions, in which case aligning functional residues may not be useful. If more distant sequences are included, we recommend using a structure-aware aligner such as MAFFT-DASH (20) or an HMM-based aligner such as HMMer's hmmalign.

SitesBLAST uses MUSCLE (version 3.8.31) with fast options (-maxiters 2 -maxmb 1000) and FastTree 2.1.11 with default settings.

**Data availability.** The code is available as part of the PaperBLAST code base (https://github.com/morgannprice/PaperBLAST). The SitesBLAST and PaperBLAST databases are updated every 2 months. The April 2022 database is archived at figshare (https://doi.org/10.6084/m9.figshare.20022590.v1). Instructions for downloading the current database are at https://github.com/morgannprice/PaperBLAST#Download.

## ACKNOWLEDGMENTS

This material by Ecosystems and Networks Integrated with Genes and Molecular Assemblies (ENIGMA; http://enigma.lbl.gov), a Science Focus Area program at Lawrence Berkeley National Laboratory, is based upon work supported by the U.S. Department of Energy, Office of Science, Office of Biological & Environmental Research under contract number DE-AC02-05CH11231.

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
