## [Reviewer comments · mSystems]

Interactive analysis of functional residues in protein families

Morgan Price and Adam Arkin

Corresponding Author(s): Morgan Price, Lawrence Berkeley National Laboratory

Review Timeline:

Submission Date:	August 1, 2022
Editorial Decision:	September 3, 2022
Revision Received:	October 4, 2022
Accepted:	October 24, 2022

Editor: Marnix Medema

Reviewer(s): Disclosure of reviewer identity is with reference to reviewer comments included in decision letter(s). The following individuals involved in review of your submission have agreed to reveal their identity: Janani Durairaj (Reviewer #1); Remi Zallot (Reviewer #2); Daniel H Haft (Reviewer #3)

Transaction Report:

DOI: <https://doi.org/10.1128/msystems.00705-22>

September 3, 2022

Dr. Morgan N Price
Lawrence Berkeley National Laboratory
Environmental Genomics and Systems Biology Division
1 Cyclotron Road
Berkeley

Re: mSystems00705-22 (Interactive analysis of functional residues in protein families)

Dear Dr. Morgan N Price:

Thank you for submitting your manuscript to mSystems. We have completed our review and I am pleased to inform you that, in principle, we expect to accept it for publication in mSystems. However, acceptance will not be final until you have adequately addressed the reviewer comments.

Preparing Revision Guidelines

Sincerely,

Marnix Medema

Editor, mSystems

Journals Department
Reviewer comments:

Reviewer #1 (Comments for the Author):

The authors describe two related tools for comparing and contrasting functional residues across homologous proteins. These could be useful for biologists interested in a particular protein as well as for manual curation of protein function. "Functional" residues are clearly defined as those annotated as ligand-binding or active sites in BioLiP, or as covalent modifications, experimentally mutated, or functional sites in Swiss-Prot. The authors estimate the coverage of their database with a random set of proteins and conclude that their tool can identify potential functional residues for half of the cases. They then describe and demonstrate the visualizations produced by their tools.

Major comments:

The motivations and advantages behind separating pairwise alignments and multiple sequence alignments (with an associated phylogenetic tree) into two different tools is not clearly explained. With the "functional positions" and "all positions" option in Sites on a Tree one can obtain pretty much the same information as in SitesBLAST (and more since you now see patterns across the whole family). The only difference is that there is no link to the relevant papers which could instead be added to Sites on a Tree.

The caption for Figure 3 is not a complete description of the figure and could be expanded to better explain what we are looking at. (Related, see comment about amino acid color bar in the minor comments)

SitesBLAST, unlike Sites on a Tree, does not seem to offer any text export options: of the alignments, functional sites, or of the associated PubMed IDs. This would be essential for proper reporting and for downstream tasks.

I submitted UniProt ID P03088 to Sites on a Tree and received an error "Invalid subjectId 6esb:1 from subject PDB:6esb:1 at /opt/apache/vhosts/genomics.lbl.gov/papers/Genomics/cgi/treeSites.cgi line 272." While the two examples on the website work as expected, submitting BT2157 (the first example) as a UniProt ID only returns a single homolog to build the alignment making it unclear how the first example alignment was produced.

Line 180 does not fully describe the page obtained after clicking on a leaf node The mentioned "links" in fact link to a large number of useful databases and tools which are not described or cited in the manuscript.

The Discussion is a bit sparse, I would expect some comments about linking additional sources of information such as structural closeness between residues, similarity of bound ligands across the family, conservation statistics across the generated alignment, available protein function and variant effect predictors, and so on. Also about how the database could be expanded in the future to close the coverage gap.

Minor comments

- In SitesBLAST, residue mismatches could also be highlighted in the list section underneath the alignment as they are difficult to spot otherwise.
- In Sites on a Tree, it would be helpful to have a colorbar depicting the amino acid color scheme used, especially for the SVG export.
- Line 190, "screenfull" depends on screen size, would be good to specify a reasonable assumption of screen size.

Reviewer #2 (Comments for the Author):

The manuscript authored by Price and Arkin, titled "Interactive analysis of functional residues in protein families" describes the justification for, the working and the example of use of two webtools: "SitesBLAST" and "Sites on a Tree".

Both webtools intended aim is to allow biologists to visualize, analyse and explore the conservation of specific residues in homologous sets of protein sequences.

"SiteBLAST" uses information available from the SwissProt and BioLip databases to identify, point at and compare residues that are proposed key for catalytic activity, or involved in binding of metabolites or ions, in the query sequence. It displays a BLAST output from an analysis for query sequence, aligned with best hits, into which key identified key residues are highlighted. The capabilities provided by this webtool are useful. The limitation of this webtool, as I see it, is based on the limited information that is available within SwissProt and BioLip, representative of the common knowledge obtained over years by scientists. In other words, in my opinion, if an input sequence is unrelated at the sequence level to any sequence that has SwissProt or BioLip information available, this webtool will unlikely provide a useful output.

"Sites on a Tree" allows to generate protein phylogenetic trees for related sequences and highlight the conservation (or lack of conservation) of specific selected residues across the sequences present in the tree. Default option builds upon "SiteBLAST", and highlight key residues, as identified present in SwissProt or BioLip, but is largely customisable, into being able to highlight any selected residues that can be of interest. Default input is a single sequence of interest, but accepts multiple input forms.

Overall, both websites appear very useful, extremely responsive and fast. Both tools seem to work very well. The outputs provided are extremely valuable, and highly customizable in for "Sites on a Tree".

The negative criticism that I would express, valid for both websites, is that the visual appeal is limited. However, a useful service is more important than a service that is visually appealing but useless or simply not working. Another point is that, at the moment, both websites are only anecdotal mentions on the PaperBLAST webpage. Both tools may benefit from having their own landing pages.

Thus, overall, both "SitesBLAST" and "Sites on a Tree" should be recommended for use to a wide audience. I can imagine biologists finding this tool very useful in various applications. I also believe that this tool could be powerful in an educational environment, with selected appropriate examples.

The manuscript that describes those tools is clearly written, and very accessible.

Reviewer #3 (Comments for the Author):

It would be good to cite PMID:12520028 as an example of resource that is aware of sites and can show how they line up. It does not provide a means to use the sites to attach annotations to proteins, only to inform the user, but that use is comparable to what SitesBLAST provides - a useful abstraction of sites information.

In this manuscript, Price and Arkin present a pair of related tools that use the same high-value input data set, and related search methods, for finding informative sequences from among all those known, and the most informative amino acid positions within those sequences, to benefit biochemistry researchers and students.

The intuitive interface for *SitesBLAST* starts with a query sequence and finds homologs with information about functionally important sites. It lets the user see at a glance which key sites have been described in publications and subsequently mined by curators and incorporated into SwissProt entries, and if those sites are the same or different. These include active sites, binding sites, cleavage sites, and modification sites. It also includes sites from BioLiP, is a database of protein-ligand interactions mined computationally from solved crystal structures and filtered for relevance. Perhaps surprisingly, BioLiP provides *SitesBLAST* with the majority of its substrate-binding and cofactor-binding sites. In just a few seconds, *SitesBLAST* helps answer questions such as “what is known about key sites in WP_044084028.1, precursor of the cofactor mycofactocin, in *Mycobacterium tuberculosis*?” or “what is the role of the selenocysteine (U) residue in WP_004074143.1”

The related tool *Sites on a Tree* provides a generalization from pairwise alignments to multiple sequence alignments. Users wishing to learn rapidly what is known about a particular protein family, such as subclass B3 metallo-beta-lactamases such as SIE-1 (WP_007683232.1), can easily see what residues have been described in the literature or seen binding ligand in crystal structures, and see which ones actually are well conserved. For SIE-1, the user sees easily that Glu-122 replaces a His residue known to bind a critical Zn ion in essentially all well-studied homologs – a finding that may be important to surveillance for new kinds of antibiotic resistance genes. *Sites on a Tree* users can simply accept default behaviors to select proteins, alignment them, filter the alignment for informative positions, root the tree, in order to see patterns of amino acid substitutions at annotated key sites that either follow or break with the topology of the tree. The advanced user, perhaps the researcher investigating structure-function relationships, can opt to upload improved versions of any of the required files.

In many ways, the tools described here resemble *PaperBLAST* in the way they accelerate access to sought-after information. An open access journal article is still limited in its true accessibility if no good search method can lead users to it. *PaperBLAST* (<https://papers.genomics.lbl.gov/cgi-bin/litSearch.cgi>) lets users start with a protein sequence, then discover through a BLAST search whether any of a million different articles mention homologs to the query by accession number or locus tag, and if so, what they call those homologs. For the biocurator building new automated rules for genome annotation pipelines, *PaperBLAST* is invaluable. *SitesBLAST* offers a similar giant step forward in access to information about sites determining function, long available in principle by painfully slow manual methods, but now available through rich and easy to read output of a high speed bioinformatics tool with a friendly and intuitive user interface.

Responses to comments from reviewer #1

> The motivations and advantages behind separating pairwise alignments and multiple sequence alignments (with an associated phylogenetic tree) into two different tools is not clearly explained.

We added a paragraph to the beginning of the Discussion to discuss the tradeoffs between the two tools. Briefly, SitesBLAST is simpler and quicker to use, while Sites on a Tree is more powerful and flexible.

> The only difference is that there is no link to the relevant papers which could instead be added to Sites on a Tree.

We have updated Sites on a Tree so that, when you click on a node or sequence in the tree to view information about that sequence, it includes links to curated information and papers (when available). For instance, most of the sequences in the BT2042 example now have links to papers. Due to these changes, Sites on a Tree now generates different identifiers for some of the sequences in that example, so Figure 2 has also changed slightly.

> The caption for Figure 3 is not a complete description of the figure and could be expanded to better explain what we are looking at.

We expanded the caption.

> SitesBLAST, unlike Sites on a Tree, does not seem to offer any text export options: of the alignments, functional sites, or of the associated PubMed IDs. This would be essential for proper reporting and for downstream tasks.

We added an option to download a tab-delimited table, with one row for each hit's functional site; each row includes the sequence of the site, the aligned region of the query, and any PubMed identifiers.

> I submitted UniProt ID P03088 to Sites on a Tree and received an error "Invalid subjectId 6esb:1 from subject PDB:6esb:1 at /opt/apache/vhosts/genomics.lbl.gov/papers/Genomics/cgi/treeSites.cgi line 272."

This has been fixed. There was a bug in how sequences from PDB with integer chains (chain "1" instead of "A") were handled. We also did some large-scale testing and did not find any other pages that fail.

> While the two examples on the website work as expected, submitting BT2157 (the first example) as a UniProt ID only returns a single homolog to build the alignment making it unclear how the first example alignment was produced.

We revised the example on the web site to describe the origin of these sequences. (This was described in our original submission, but not on the web site.)

> Line 180 does not fully describe the page obtained after clicking on a leaf node. The mentioned "links" in fact link to a large number of useful databases and tools which are not described or cited in the manuscript.

We modified the text to explain that the links are to other sequence analysis tools. We thought that explaining what all these other tools were would interrupt the flow of the manuscript, so we do not describe them further.

> The Discussion is a bit sparse, I would expect some comments about linking additional sources of information such as structural closeness between residues, similarity of bound ligands across the family, conservation statistics across the generated alignment, available protein function and variant effect predictors, and so on.

We added some text in the Discussion about the Conserved Domains Database (CDD), which was requested by another reviewer. We also moved the brief comparison to firestar from the Results to this part of the Discussion.

We agree that it would be helpful to show if the functional sites are nearby in the protein structure, and we hope to add this capability in the future.

We felt that the other automated analyses that the reviewer mentioned are less relevant to our manuscript, which focuses on manual analyses of functional sites. The Discussion does mention that Sites on a Tree might be useful for visualizing the predicted functional sites from automated tools (i.e., Laurie and Jackson 2005).

> Also about how the database could be expanded in the future to close the coverage gap.

As mentioned in the Discussion, there's often more information in the papers (that discuss the structure or that are referenced by the curators of SwissProt) than in the databases. But we think the main challenge is that many sequences are not similar to a protein with known functional residues (or, they are very-distantly related to such a protein, and the functional residues have moved). So, we don't know how to close the coverage gap. We added a sentence to the Discussion to emphasize that for many protein families, there is no experimental evidence as to their functional sites. Also, we do mention tools for predicting functional residue (i.e. Laurie and Jackson 2005), but we don't necessarily see this as a solution.

> In SitesBLAST, residue mismatches could also be highlighted in the list section underneath the alignment as they are difficult to spot otherwise.

To make these mismatches more prominent, they are now shown with a red inequality symbol.

> In Sites on a Tree, it would be helpful to have a colorbar depicting the amino acid color scheme used, especially for the SVG export.

We have added a legend at the bottom for the amino acid coloring.

> Line 190, "screenfull" depends on screen size, would be good to specify a reasonable assumption of screen size.

We reworded this to emphasize that this reflects our experience. Users of other computers might have a different experience.

Responses to comments from reviewer #2

> The limitation of this webtool, as I see it, is based on the limited information that is available within SwissProt and BioLip, representative of the common knowledge obtained over years by scientists. In other words, in my opinion, if an input sequence is unrelated at the sequence level to any sequence that has SwissProt or BioLip information available, this webtool will unlikely provide a useful output.

Mostly we agree with this comment. But if a protein structure is available, it is sometimes possible to identify functional sites (using other tools) and then to visualize them with Sites on a Tree. We do show an example of this (Figure 3).

> The negative criticism that I would express, valid for both websites, is that the visual appeal is limited. However, a useful service is more important than a service that is visually appealing but useless or simply not working.

We revised the look of the alignments in SitesBLAST. We hope this is an improvement.

> Another point is that, at the moment, both websites are only anecdotal mentions on the PaperBLAST webpage. Both tools may benefit from having their own landing pages.

Every PaperBLAST results page links to the SitesBLAST page for that query. We also modified how SitesBLAST and Sites on a Tree link to each other and to PaperBLAST. We hope this will make the tools more easily accessible.

Responses to comments from reviewer #3

> It would be good to cite PMID:12520028 as an example of resource that is aware of sites and can show how they line up. It does not provide a means to use the sites to attach annotations to proteins, only to inform the user, but that use is comparable to what SitesBLAST provides - a useful abstraction of sites information.

We added a paragraph to the Results to compare the coverage of SitesBLAST's database to that of CDD (the tool described by the reference). Overall, 49% of prokaryotic proteins have hits from SitesBLAST, versus 40% for CDD.

We also added a paragraph to the Discussion to compare the capabilities of SitesBLAST and CDD. (We also moved the brief comparison to firestar from the Results to the Discussion.)

October 24, 2022

Mx. Morgan N Price
Lawrence Berkeley National Laboratory
Environmental Genomics and Systems Biology Division
1 Cyclotron Road
Berkeley

Re: mSystems00705-22R1 (Interactive analysis of functional residues in protein families)

Dear Dr. Morgan N Price:

Your revised manuscript has now been seen by the same three reviewers, who are all satisfied with the changes made.

Your manuscript has been accepted, and I am forwarding it to the ASM Journals Department for publication. For your reference, ASM Journals' address is given below. Before it can be scheduled for publication, your manuscript will be checked by the mSystems production staff to make sure that all elements meet the technical requirements for publication. They will contact you if anything needs to be revised before copyediting and production can begin. Otherwise, you will be notified when your proofs are ready to be viewed.

Publication Fees:

If you would like to submit a potential Featured Image, please email a file and a short legend to mSystems@asmusa.org. Please note that we can only consider images that (i) the authors created or own and (ii) have not been previously published. By submitting, you agree that the image can be used under the same terms as the published article. File requirements: square dimensions (4" x 4"), 300 dpi resolution, RGB colorspace, TIF file format.

We recognize that the video files can become quite large, and so to avoid quality loss ASM suggests sending the video file via <https://www.wetransfer.com/>. When you have a final version of the video and the still ready to share, please send it to mSystems staff at mSystems@asmusa.org.

Sincerely,

Marnix Medema
Editor, mSystems

Journals Department
E-mail: mSystems@asmusa.org